
# Application of time-of-flight aerosol mass spectrometry for the real-time measurement of particle phase organic peroxides: An online redox derivatization – aerosol mass spectrometer (ORD-AMS)

Marcel Weloe[1] and Thorsten Hoffmann[1]

[1]Department of Chemistry, Johannes Gutenberg University, Mainz, 55128, Germany

*Correspondence to*: Thorsten Hoffmann (t.hoffmann@uni-mainz.de)

**Abstract.** Aerosol mass spectrometers (AMS) are frequently applied in atmospheric aerosol research in connection with climate, environmental or health related projects. This is also true for the measurement of the organic fraction of particulate matter, still the least understood group of components contributing to atmospheric aerosols. While quantification of the organic/inorganic aerosol fractions is feasible, more detailed information about individual organic compounds or compound classes can usually not be provided by AMS measurements. In this study we present a new method to detected organic peroxides in the particle phase in real-time using an AMS. Peroxides (ROOR') are of high interest to the atmospheric aerosol community due to their potentially high mass contribution to SOA, their important role in new particle formation and their function as 'reactive oxygen species' in aerosol-health-related topics. To do so, supersaturated gaseous triphenylphosphine (TPP) was continuously mixed with the aerosol flow of interest in a condensation/reaction volume in front of the AMS inlet. The formed triphenylphosphine oxide (TPPO) from the peroxide/TPP reaction was then detected by an aerosol mass spectrometer (AMS), enabling the quantitative determination of peroxide with a time resolution of one to two minutes. The method was tested with freshly formed and aged biogenic VOC/ozone SOA as well as in a short proof-of-principle study with ambient aerosol.

## 1 Introduction

Atmospheric aerosols attract attention in atmospheric research owing to their effects on climate and human health. The climate relevance of aerosol particles arises from their influence on the transmissivity of the atmosphere for solar radiation and on cloud properties (McNeill, 2017). Secondary organic aerosols (SOAs) are a major contributor to tropospheric particulate matter (Hallquist et al., 2009; Poschl, 2005). One highly relevant class of compounds which is assumed to play an essential role in SOA formation, including participation in new particle formation, are organic peroxides (Bonn et al., 2004; Mochida et al., 2006; Reinnig et al., 2008; Reinnig et al., 2009; Tobias et al., 2000; Tobias and Ziemann, 2002; Zhou et al., 2018; Bianchi et al., 2016; Bianchi et al., 2017; Bianchi et al., 2019; Ziemann, 2002, 2003; Kahnt et al., 2018; Zhang et al., 2017; Lee et al., 2019). Beside their significant contribution to SOA mass (Docherty et al., 2005b; Ziemann and Atkinson, 2012), particle phase peroxides are especially interesting due to their often very low saturation vapor pressure, particularly when they contain several hydroperoxy groups induced by autoxidation (Troestl et al., 2016; Krapf et al., 2016). However, also their chemical properties such as their oxidizing character (Claflin et al., 2018) or their function as indicator for the oxidizing capacity of the troposphere (Thompson, 1992; Lelieveld et al., 2008) create special interest in the analysis of organic peroxides. Finally, peroxide functionalities are likely to play an important role in health related effects of SOA, since organic peroxides are believed to contribute to the formation of reactive organic species (ROS) by transporting oxidants on/in particles into the respiratory system (Verma et al., 2015; Wragg et al., 2016; Fuller et al., 2014; Steimer et al., 2017; Tong et al., 2018).

The fact that organic peroxides are oxidizing agents is often used in their analysis. One of the oldest and most frequently applied techniques to measure and quantify organic peroxides is iodometry (Mertes et al., 2012b, 2012b; Docherty et al.,



2005a; Mutzel et al., 2013; Zhao et al., 2018). Alternatively, triphenylphosphine (TPP) can be used as reacting agent. TPP rapidly reacts with peroxides to triphenylphosphine oxide (TPPO) (Docherty et al., 2004; Ruiz et al., 2001), which can be detected spectrometrically (Porter et al., 1979). The main application of the TPP/TPPO-method is the determination of hydroperoxides from the autoxidation of oils and fats (Gotoh et al., 2011; Nakamura and Maeda, 1991; Wang et al., 2016). Compared to the iodometric peroxide determination the TPP/TPPO-method is not sensitive to moisture and the reaction with oxygen is much slower.

Triphenylphosphine (TPP)
MW 262 g mol$^{-1}$

Triphenylphosphine oxide (TPPO)
MW 278 g mol$^{-1}$


Another technique that has been used for the analysis of organic peroxides is mass spectrometry (Krawczyk and Baj, 2014), for example for the measurement of SOA related organic peroxides by direct infusion of liquid samples into an atmospheric pressure chemical ionization–tandem mass spectrometer (Zhou et al., 2018). However, all the above mentioned techniques are offline methods relying on filter sampling techniques, which have disadvantages such as the evaporation of semi-volatile
species or chemical alterations of labile compounds during sampling. Obviously also the temporal resolution of offline methods is limited. Therefore, online methods for organic peroxide measurements would be beneficial.

One of the main online techniques that is used in the field of aerosol research is the aerosol mass spectrometer (AMS), which is commercially available from Aerodyne Research Inc. (Billerica, MA, USA). Aerosol particles are introduced into a vacuum chamber by means of an aerodynamic lens that focuses the sampled aerosol particles to a beam (Jayne et al., 2000). The particle
beam is directed onto an evaporation unit, which is heated to 600 °C, ideally volatilizing the entire particle mass. The evaporated compounds are subsequently ionized by electron ionization (EI) at 70 eV and analyzed by MS (Pratt and Prather, 2012a, 2012b; Jayne et al., 2000). The application of EI allows the analysis of a wide range of chemical species and provides robust quantitative information on various non-refractory ambient aerosol components since the ionization efficiency of EI is much less analyte and matrix dependent than, for example, chemical ionization techniques (Allan et al., 2003b; Allan et al.,
2003a; Canagaratna et al., 2007a). However, EI induces a high degree of fragmentation of organic compounds due to excess internal energy imparted during the ionization processes. Usually, these fragmentation steps impede the identification and quantification of individual particle phase organics, thus leading to a reduced level of chemical information (Zhang et al., 2011). Therefore, recent mass spectrometric instrumental developments for organic aerosol analysis focus on softer ionization techniques such as atmospheric pressure chemical ionization (APCI-MS) (Vogel et al., 2013b; Vogel et al., 2013a;
Brüggemann et al., 2014; Warscheid and Hoffmann, 2001a, 2001b), flowing atmospheric-pressure afterglow mass spectrometry (AeroFAPA-MS) (Brueggemann et al., 2015; Brueggemann et al., 2016; Brueggemann et al., 2017) or extractive electrospray ionization (EESI-MS) (Gallimore et al., 2017). However, while these mass spectrometric techniques are applicable for a wide spectrum of organic compounds and functionalities, for example carboxylic acids (Hallquist et al., 2009; Yatavelli et al., 2015), unambiguous identification of organic peroxides using mass spectrometric techniques is still
challenging, since in contrast to other functionalities the molecular ions usually fragment resulting from the cleavage of the peroxide bond and/or further fragmentation (Reinnig et al., 2009; Guascito et al., 2015).

Here, we have combined the concept of selective peroxide determination by a suitable redox-reaction (the TPP – TPPO system) and to use this reaction as an online derivatization method to monitor the educts and products with an aerosol mass spectrometer



(online redox derivatization-AMS, ORD-AMS). The development and application of the method presented in this paper is

essentially a proof-of-principle study, however, the results clearly show the potential of online derivatization as an extension of regular AMS measurements, in this case for the understanding of particle phase organic peroxides. Similar experimental setups with different derivatization agents are also conceivable in the future to further extent the usability of AMS systems.

## 2 Experimental

### 2.1 Online redox derivatization

Figure 1 shows a schematic diagram of the experimental setup for the AMS online measurements in the ORD-AMS mode. Aerosol particles, produced either by an aerosol generator (nebulizer) or within a photochemical reaction chamber, are introduced into a 500 mL glass tube (35 cm length, max. diameter 4.5 cm), the condensation/reaction volume. Here, the aerosol is mixed with the gaseous reactant (TPP, 99%, Acros Organics BVBA, Geel, Belgium), which is supplied from a Sinclair-Lamer setup (see Figure 1, blue triangle). The TPP source consists of a two-necked round bottom glass filled with TPP and is

heated to a temperature of 90 °C. Nitrogen is used as a carrier gas and deliver the TPP to the condensation/reaction volume, where TPP condensed on the available particle surface. After passing the condensation/reaction volume the aerosol is led through a denuder (inner tube: diameter = 2 cm, length 40 cm, outer tube: diameter 13.5 cm, length 40 cm) filled with active charcoal to remove gas phase TPP and finally introduced into the mass spectrometer and a condensation particle counter (CPC, PortaCountPlus, TSI In., Shoreview, MN, USA).

### 2.2 ToF-AMS

The aerosol particles were introduced into the Time-of-Flight (ToF)-AMS (Aerodyne Research Inc., Billerica, MA, USA) at a sampling flow rate of 1.23 mL s$^{-1}$. An aerodynamic lens system focuses the particles in a size range of ~35−1000 nm into the vacuum system. While most of the remaining gas phase is removed, the particle beam is transmitted into the particle sizing chamber. The ToF-AMS was either operated in the Mass Spectrum (MS) or the Particle Time-of-Flight (PToF) mode. The

first mode provides information of the overall composition of the total aerosol mass without any information about the particle size, the second mode results in size-segregated analysis of the particle composition. For the latter one, a chopper allows the transmission of only a small part of the aerosol into the PToF chamber. Due to increasing inertia with larger particle size, the particle package is separated by their time of flight and the size distribution based on the mean vacuum aerodynamic diameter of the aerosol can be determined. In this PToF-mode, the EI-ToF-MS functions as detector for the particle time of flight. To

determine the concentration in the MS-mode, the beam pass the chopper alternately blocked or unblocked for background corrections. After passing the particle sizing chamber, the aerosol particles are vaporized on a heated porous tungsten surface at 600 °C followed by immediate ionization in electron impact (EI) mode at 70 eV. The generated ions are extracted by an orthogonal extractor into the time ToF-MS, which was operated in the V-mode to give a higher sensitivity. The MS spectra acquisition was performed in the positive ion mode. The AMS data were processed by the Igor (Version Pro 6.37) based

programs "SQUIRREL 1.57 G" (unit mass resolution) and "PIKA 1.16H" (high resolution) (DeCarlo et al., 2006; Canagaratna et al., 2007b).

### 2.3 Physical aerosol characterization and generation

A Scanning Mobility Particle Sizer (SMPS, Grimm Aerosol Technik, Ainring, Germany) was used to measure aerosol number size distributions of the incoming aerosol. Beside the measurement of the aerosol size distribution the instrument was used to

generate monodisperse aerosol for instrument calibration. The EI-spectra of the derivatization agent TPP and its reaction product with peroxides TPPO (> 98 %, Merck KGaA, Darmstadt, Germany) were recorded using the nebulizer setup (Fig. 1). Pure TPP and TPPO particles were produced from solutions in methanol (Optima® LC/MS grade, Fisher scientific,



Loughborough, UK) and *n*-heptane (HPLC grade, Carl Roth GmbH + Co. KG, Karlsruhe, Germany), respectively. For these experiments the particles were size selected using the DMA in a setup as described in SI.

### 2.4 Reaction chamber as SOA generator

For the proof-of-principle studies on SOA a laboratory scale reaction chamber in CSTR mode was used. Synthetic air was supplied with a total flow of 7.6 L min$^{-1}$ into the 100 L chamber. The total flow was composed of a gas stream through a diffusion test gas source (Thorenz et al., 2012) (0.600 L min$^{-1}$), where the VOC of interest was added to the synthetic air supply. A second gas stream (2.6 L min$^{-1}$) was humidified using a gas washing bottle and third gas flow was led through an ozone generator (Dasibi Environmental Corp. Model 1008 RS O3 analyzer, Clendale, CA, USA, 2.7 L min$^{-1}$). To add seed particles an aqueous solution of ammonium sulfate (ca 0.5 g/L, > 99.5%, Merck KGaA, Darmstadt, Germany) were nebulized using technical nitrogen and introduced into the chamber (1.7 L min$^{-1}$). Temperature and relative humidity were monitored by a thermo/hygrometer (Amarell, ama-digit ad 910 h, Kreuzwertheim, Germany). The chamber was set under slight overpressure to prevent entry of laboratory air into the setup. The reaction chamber was wrapped with aluminum foil to avoid photochemical reactions. For the ozonolysis of α-pinene, the chamber was first equilibrated with the terpene before the ozone generator was switched on. For the ß-pinene (99%, Sigma Aldrich Chemie GmbH, Steinheim, Germany) experiments, the order of the addition of the individual compounds was reversed, i.e. ozone was added before the terpene. The ozone concentration was estimated to be about 1 ppm, the α-pinene (> 99%, Fluka, Seelze, Germany) concentration was set to 290 ppbv and the β-pinene concentration to 380 ppbv. To remove gaseous organic compounds the SOA formed in the reaction chamber was led through two diffusion denuders filled with activated charcoal.

## 3 Results and discussion

### 3.1 EI mass spectra

Figure 2 shows the obtained mass spectra of TPPO and TPP. The EI spectra of the two reference compounds measured by the AMS are very similar to the spectra measured by regular MS setup (Williams et al., 1968; Stein and Slawson, 1963). The base peak for TPPO is at *m/z* 277 resulting from a hydrogen abstraction ([M-1]$^+$) from the molecular ion (*MW* (TPPO) = 278 g mol$^{-1}$). The second largest peak is *m/z* 77 formed from the abstraction of a phenyl cation. The base peak for TPP is located at *m/z* 183 originating from the abstraction of two hydrogens and a phenyl group, probably a result of an aromatic stabilization of the resulting dibenzophosphole cation (Williams et al., 1968). However, also the molecular ion of TPP at *m/z* 262 is visible. More important is the small but still observable [M-1]$^+$ ion of TPPO in the TPP spectra. Since the peroxide determination is based on the measurement of TPPO, this background signal hat to be taken into account (see below).

### 3.2 Quantification

The mass concentration $c_m$ of a single compound measured by an AMS can be expressed as (Canagaratna et al., 2007b):

$$c_m = \frac{10^{12} \cdot MW\,(compound)}{IE\,(compound) \cdot Q \cdot N_A} \cdot \sum_{all,i} I_{compound,i}$$

where *MW* and *IE* is the molar weight and the ionization efficiency of an individual compound. The ionization efficiency (IE) is a dimensionless quantity that describes the ionization and detection efficiency and is defined as the ratio of the ions detected to the parent molecules vaporized. $Q$ is the flowrate into the AMS, $N_A$ the Avogadro's number and $10^{12}$ a unit conversion factor. $I_{compound,i}$ is the ion rate of an ion $i$ formed by ionization and fragmentation of a compound. Since the fragmentation pattern can be assumed to be robust, it is possibly to use the ion yield of a single ion $I_{m/z,i}$ to calculate the concentration. The





molar concentration $c_n$ of peroxides is according to the reaction equation the same as the concentration of TPPO and can be calculated from the signal at $m/z$ 277 as:

$$c_n(TPPO) = \frac{10^{12}}{IE\left(TPPO, \frac{m}{z}i\right) \cdot Q \cdot N_A} \cdot I_{\frac{m}{z}i}$$


$$c_n(TPPO) = \frac{10^{12}}{IE\left(TPPO, \frac{m}{z}277\right) \cdot Q \cdot N_A} \cdot I_{\frac{m}{z}277}$$

$$c_n(TPP) = \frac{10^{12}}{IE\left(TPPO, \frac{m}{z}262\right) \cdot Q \cdot N_A} \cdot I_{\frac{m}{z}262}$$

The experimental procedure to estimate IE is discussed in more detail in SI.

### 3.3 Background correction

As mentioned above also TPP particles as well as pure ammonium sulfate aerosol particles passing the TPP-condensation/reaction chamber show a small but non-negligible TPPO signal ($m/z$ 277). This is either a consequence of TPPO impurities in the TPP itself or due to traces of peroxides in the solvents used to generate the aerosol particles when the nebulizer was used. Consequently, a background correction procedure has to be introduced to account for these effects. To do so, pure AS aerosol particles were investigated with different initial ammonium sulfate concentrations with the ORD-AMS setup as
described above. These experiments showed a constant background contribution of 1,222 (+/-0.1) % TPPO from the conversion of TPP ($m/z$ 262 => $C_n$ (detected TPP)). Therefore, the concentration of the analyte of interest ($C_n$ (peroxide)) was calculated as follows:

$C_n$(backgr. TPPO) $= 0,01215 \cdot C_n$(detected TPP)     (1)
$C_n$(produced TPPO) $= C_n$(detected TPPO) - $C_n$(backgr. TPPO)     (2)
$C_n$(peroxide) $= C_n$(produced TPPO)     (3)

### 3.4 SOA experiments

The ozonolysis of α- or ß-pinene was performed in the presence of ammonium sulfate (AS) seed particles and the results of a typical chamber run with α-pinene is shown in Figure 3. First, AS seed particles were injected into the chamber. After a certain
period of time, when the particle concentration became constant, the biogenic VOC was introduced into the chamber (e.g. after 34 min in the run shown in Fig. 3). Following an initial period to monitor background peroxide concentrations in the AS seed aerosol / VOC system, ozone was introduced (e.g. after 135 min in the α-pinene run shown in Fig. 3 (green area)). Periods, when TPP was not added, are marked in grey and were used to determine the dilution factor and the concentration of ammonium sulfate and SOA by the AMS. As can be seen in the figure, the concentrations of both TPPO and TPP rose quickly
during the first 20-25 min after the ozone addition (Fig. 3a), followed by a slow decrease towards the end of the experimental run. In fact, peroxides were only detected when SOA formation started, which is indicated by the time series of peroxide concentration (Fig. 3b, black line) and the total aerosol volume concentration (Fig. 3b, red line). Figure 3c shows the temporal behavior of the relative contribution of peroxides to total SOA. The course of the peroxide amount per particle volume (Figure 3c) shows an increase within the first about 20 min, followed by a decrease towards the later stage of the experiment. Such
concentration profiles of the organic peroxides are indicative of consecutive reactions, in which the peroxides are formed form the initial ozone/alkene reaction (A + B → C), followed by a further reaction of the peroxides to non-peroxidic products (C → D). Indeed, several studies suggest that organic peroxides are instantly formed in the VOC-ozone reaction (Bianchi et al., 2019;


Mertes et al., 2012a) and the rapid increase of the peroxide signal as shown in Figure 3b confirms these studies. However, since the O-O bond is a relatively weak bond, organic peroxides tend to decompose or react with other aerosol components.

Possible reactions to non-peroxidic products could be reactions with aldehydes forming carboxylic acids or hydrolysis followed by the loss of $H_2O_2$ (Zhao et al., 2018; Claflin et al., 2018; Epstein et al., 2014; Claflin et al., 2018). In any case is the temporal evolution of the peroxide concentration (i.e. the formation/decline profile) indicative of consecutive reactions which demonstrates the intermediate character of the peroxides within the VOC oxidation system.

As can be seen in Figure 3c, the maximum contribution of peroxides to total SOA-AS was observed after about 20 minutes and was nearly 70 %. During the last 80 min of the experiment, the SOA concentration was about 58 $\mu m^3$ $cm^{-3}$ (69 $\mu g$ $m^{-3}$; 0,28 $\mu mol$ $m^{-3}$, SOA density 1,2 g $cm^{-3}$), assuming an average molar mass of the SOA components of 250 g $mol^{-1}$. During this time the measured peroxide concentration was about 0,047 $\mu mol$ $m^{-3}$. Zieman and coworkers (Docherty et al. 2005) proposed an average molar mass of 300 g $mol^{-1}$ for SOA-peroxides. Based on this value the average peroxide mass concentration was

14,1 $\mu g$ $m^{-3}$, assuming the same density of peroxides and other SOA components (1,2 g $cm^{-3}$). Thus the average peroxide/SOA yield within this period is estimated to be about 20 % (± 4%). Actually, this value is very close to the value of 22 % (Epstein et al. 2014) or the value of 21 % (Li et al. 2016). Mertes and coworker (Mertes et al. 2012) observed 34, 17 and 12 % for 15-35 min, 4 hours and 6 hours filter samples, respectively. Actually, considering the strong influence of sampling time and duration on the peroxide determination, these values are in good agreement to each other.


Figure 4 shows the particle size distribution measured by the SMPS system before (black) and within the first 10 minutes after (red) ozone was introduced into the reaction chamber. Despite the presence of AS seed particles new particle formation from nucleation of organic compounds was observed, resulting in a bimodal size distribution in the beginning of the experiment. Actually, this observation is not surprising due to the high new particle formation potential of biogenic VOCs as well as the

relatively high concentrations of the reactants. The same behavior was also visible in the AMS data. One of the main advantages of the AMS is the possibility to determine the size distributions of individual compounds. If the $m/z$ are characteristic for an aerosol component, such as $m/z$ 30 and 46 for nitrate for example, the size distribution of these components can be obtained. Figure 5 shows the $m/z$ values as a function of particle size for TPP ($m/z$ 262 (blue)), TPPO ($m/z$ 277 (black)), SOA ($m/z$ 43 (green)) and sulfate (red) for the first 10 minutes after the ozone was introduced, all normalized to the respective

maximum value. Although the determined size distributions by SMPS and AMS are not identical, likely due to scan time effects, differences in aerodynamic and mobility diameters as well as differences in the averaging procedure, the bimodal size distribution of pure organic particles and pre-existing AS containing particles is also visible in the AMS data. While the occurance of organic nucleation simultaneously with condensational growth resulting in two externally mixed particle populations is not surprising, a more quantitative description of the size distribution of selected compound groups is

challenging. As can be seen in Fig. 5, the size profiles of SOA and TPPO are very similar with a clear preference to the smaller particles sizes. Beside the obvious explanation of the existence of a pure organic particle population formed by homogeneous nucleation, an additional explanation is a size dependent condensation of the organics also on the inorganic seed particles. A unique characteristic of condensational growth is that it is proportional to the Fuchs-corrected surface area of particles and not the particle volume ("surface limited" vs "volume limited" growth). Because the surface area to volume ratio of particles

increases as their diameter decreases, condensation tends to have a larger effect on smaller particles. Therefore, after condensational growth of AS seed particles the final distribution the condensed organics can be expected to favor the smaller particles (Donahue et al., 2014; Riipinen et al., 2011). Therefore, both nucleation of organics and condensational growth will contribute to the size distribution shown in Fig. 5. Nevertheless, the results clearly show that the formation of peroxidic compounds is directly connected to SOA formation. More difficult to explain is the quite similar size distribution of TPP and

ammonium sulfate seed aerosol, since also TPP moves to the particle phase via condensation. One possible explanation could



be attractive forces between the nonpolar TPP molecules and AS, similar to a 'salting in' effect (Hyde et al., 2017). However, possibly also other effects, such as the high supersaturation of TPP in the condensation/reaction volume, might be considered to explain the observations.

Figure 6 a-d show the size distributions of $m/z$ 277 (black), 262 (blue), 43 (green) and sulfate (red) for the periods before (top) and during the last 80 min of the ozonolysis (bottom) with (right) and without (left) added TPP. Again, each of the individual ratios are normalized to its respective maximum. $M/z$ 262 corresponds to the detected TPP, $m/z$ 277 to TPPO and $m/z$ 43 to SOA species. Figure 6a shows the size distribution of pure AS particles. As can be seen by comparison of 6a and b, the diameter of sulfate remained almost constant due to the condensation of TPP, if anything then the diameter slightly decreased. Generally,

the vacuum-aerodynamic diameter is depending on the particle density (DeCarlo et al., 2004). The larger the density, the larger is $d_{va}$ for particles with the same physical diameter. The condensation of TPP (density 1,1 g cm⁻³) will lower the particle density (density AS 1,77 g cm⁻³), which compensates the increase of the physical diameter. Nevertheless, the condensation of SOA material finally led to an increase of the sulfate diameter (Fig. 6c). In Fig. 6d the size distributions of the four different components is shown. First to notice is again the clear overlap of $m/z$ 277 and $m/z$ 43, demonstrating the simultaneous presence

of TPPO and SOA species in the particle phase. The slight shift in the size distribution to larger sizes for SOA ($m/z$ 43) and TPPO is likely a result of a slightly faster growth of the pure organic particles formed by nucleation as discussed above compared to the ammonium sulfate containing particles.

A second biogenic VOC, ß-pinene, was investigated with the online redox derivatisation AMS system, this time with a reverse

introduction of the VOC and the oxidant to examine ozone as a direct oxidant for TPP. Again, the ozonolysis of β-pinene was performed in the presence of AS seed aerosol. Figure 7 shows the temporal evolution of the peroxide concentration together with the total aerosol volume concentration determined with the SMPS. As in the case of α-pinene, the peroxide concentration increased rapidly after the injection of β-pinene, once again demonstrating the direct connection of SOA and peroxide formation via the detection of TPPO. The comparison of the profiles of SOA (red) and peroxide (black) demonstrates the very

rapid peroxide formation followed by the decreasing contribution of peroxides to the particle phase in the later stages of the experiment as a consequence of consecutive reactions of the peroxides to non-peroxidic products. The strong decrease of total SOA after 240 min is induced by the end of ß-pinene addition to the reaction system. The observed shift between the peroxide and SOA concentration verifies the concept of organic peroxides as reaction intermediates, which contribute especially to the composition of fresh SOA and convert to non-peroxidic products in aged SOA particles. This is also clearly visible in the

bottom part of Figure 7, which shows the relative contribution of peroxides to the total particle volume in the course of the experiment. However, most important for the overall method evaluation are the very low and rather constant TPPO concentrations in the period between the ozone and ß-pinene addition (between 86 min and 180 min) when just AS particles were introduced into the OD-AMS system, a result that clearly excludes ozone as a direct oxidant for TPP.

### 3.5 Ambient air measurements

Finally, the ORD-AMS method was tested with ambient air directly outside the chemistry building located at the campus of the university in Mainz (49.9915928 N, 8.2312864 E). The measurements were performed between May 4th (starting time 9:30 am) and May 9th 2018 (5:30 pm). Meteorological data were measured and provided by the Institute of Physics, also located at the campus of the university. During the measurement period the air masses came from north, northeast or east and air mass composition can be expected to be especially influenced by the Frankfurt Rhine-Main Metropolitan Region with ca 5.8 million

inhabitants. Figure S4 in the SI show 72 h backwards trajectories of the air masses (every 24 hours) from May 4th to May 10th 2018. The blank value for the seeded SOA experiments was used for the background correction and values below the quantification limit were set to zero. Figure 8a shows the concentration ratios of TPPO to total TPP during the measurement



period and Figure 8b the course of the particle phase peroxide concentration. Actually, the smallest values of detected TPPO to total TPP shown in Fig. 8a lie very close to the estimated blank value level (red line). This indicates that blank values of

ambient air measurements and the blank value from the seeded SOA experiments are almost identical. Values of the peroxide concentration above the limit of quantification (LOQ) (blue line) were only obtained during weekdays (May 5th Saturday; May 6th Sunday) and at daytime. The enhanced concentration during weekdays and during the day indicates local sources of peroxides in combination with photochemistry, an observation which is in agreement with result from previous peroxide measurements (Hua et al., 2008)(Guo et al., 2014). However, due to highly variable emission sources directly outside the

chemistry building the selected measurement site was certainly not ideally for general studies about particle phase peroxide chemistry. Nevertheless, this short proof-of-principle study demonstrates that peroxides in atmospheric aerosols can be detected by the new ORD-AMS system.

## 4 Conclusions

The real-time detection of peroxides by adding gaseous TPP to aerosol particles was successfully realized using an aerosol

mass spectrometer in combination with a specifically designed online derivatization unit. The ozonolysis of α-and β-pinene in the presence of ammonium sulfate seed aerosol was used to demonstrate the potential of the analytical procedure. The results clearly show the conversion of TPP to TPPO and hence the presence of organic peroxide in biogenic SOA. Especially in the early phase of SOA formation the contribution is of peroxides is exceptionally high (up to ca. 80% v/v). However, the time-resolved measurements also show a rapid further reaction of the peroxides to non-peroxidic compounds indicative of

consecutive reactions of the peroxides. The developed instrumental setup can also be used to receive particle size-resolved information about particle phase peroxides. Again, the important role of organic peroxides in the early stages of particle formation became evident, likely induced by very low volatile multifunctional organic peroxides (HOMs). Finally, the new online redox derivatization method was successfully tested with ambient air outside of the laboratory. The results presented here represent a first attempt to better characterize the role of organic peroxides in atmospheric aerosols. Future laboratory and

field studies are certainly needed to better understand the chemistry of peroxides in organic aerosols.

*Data availability.* The data obtained by ORD-AMS measurements from the SOA experiments are available on request from Marcel Weloe.

*Author contributions.* MW and TH designed the study. MW performed the experiments to develop ORD-AMS and performed the laboratory work, calibration as well as the data analysis. MW and TH examined the results, provided input on the data analysis and interpretation and prepared the manuscript.

*Competing interests.* The authors declare that they have no conflict of interest.

*Financial support.* This work was supported by the DFG, grant number HO 1748/16-1, 'Development and application of a real-time method for the determination of selected compound classes (especially organic peroxides) in atmospheric aerosol particles using an aerosol mass spectrometer (AMS)'.


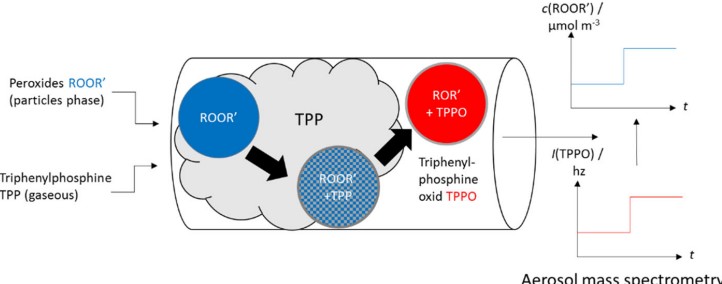

**Graphical abstract.**

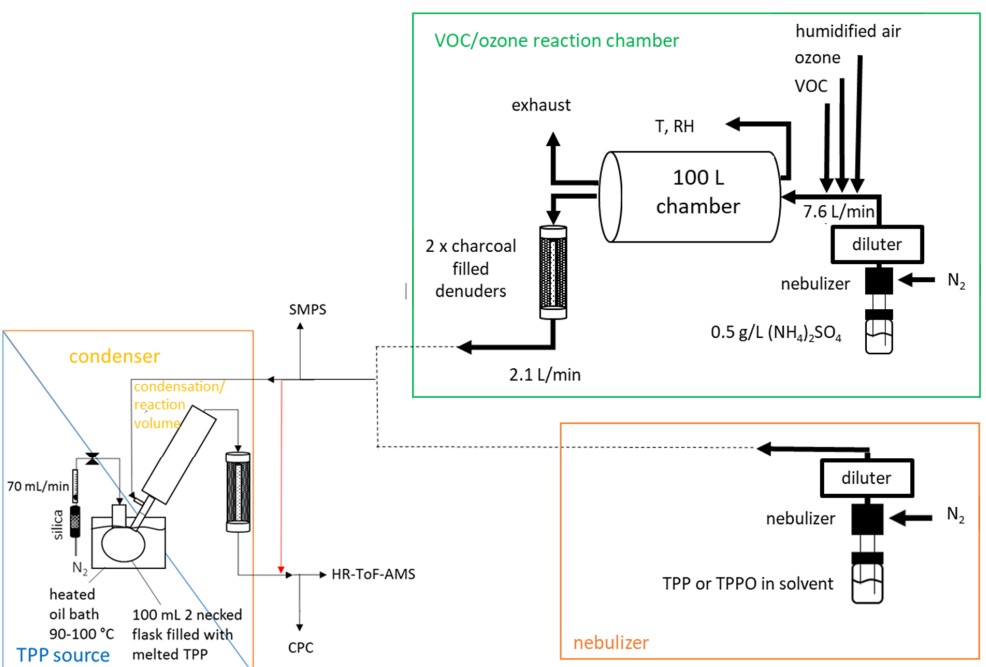

**Figure 1:** Experimental set up: The test aerosol is either generated with a nebulizer or a 100 L reaction chamber via VOC/O₃ reaction, usually also introducing AS seed particles. The test aerosol is then led to the aerosol mass spectrometer, either directly (red arrow) or through the online derivatization unit in which the test aerosol is mixed with a nitrogen stream containing supersaturated gaseous TPP. Remaining gaseous TPP is removed by a charcoal filled denuder after the condensation/reaction volume. Most of the time the aerosol is characterized by the HR-ToF-AMS, a SMPS system and a CPC.






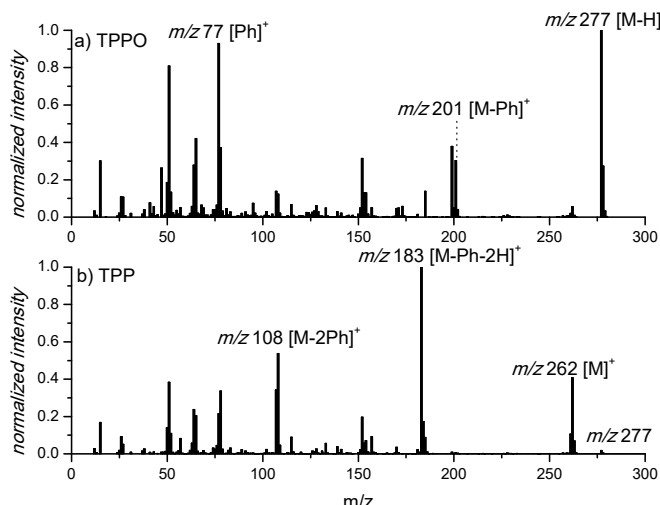

**Figure 2:** Mass spectra of a) triphenylphosphine oxide (TPPO) and b) triphenylphosphine (TPP).

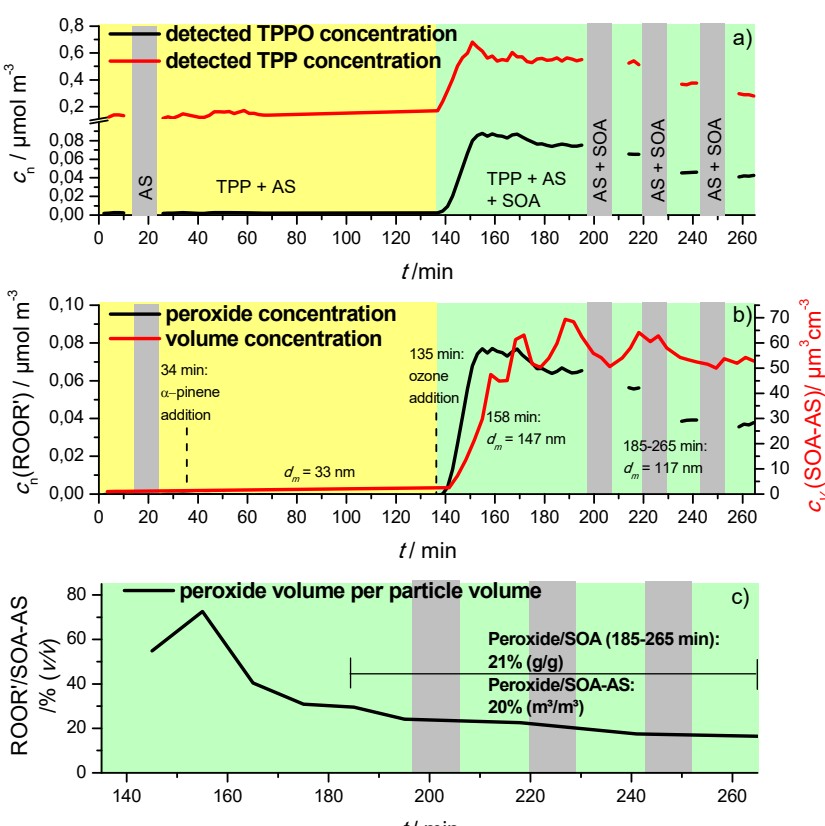

**Figure 3:** Time series of a) detected TPPO (black) and detected TPP (red), of b) peroxide concentration (black) after background correction and particle volume concentration of the SOA-AS aerosol (red) and c) relative contribution of peroxide to the total aerosol volume (black, time resolution 10 min). Periods without addition of TPP are marked with a grey background, the period before the addition of ozone is marked in yellow and during the ozonolysis in green. α-pinene was continuously added after 34 min and ozone after 135 min.





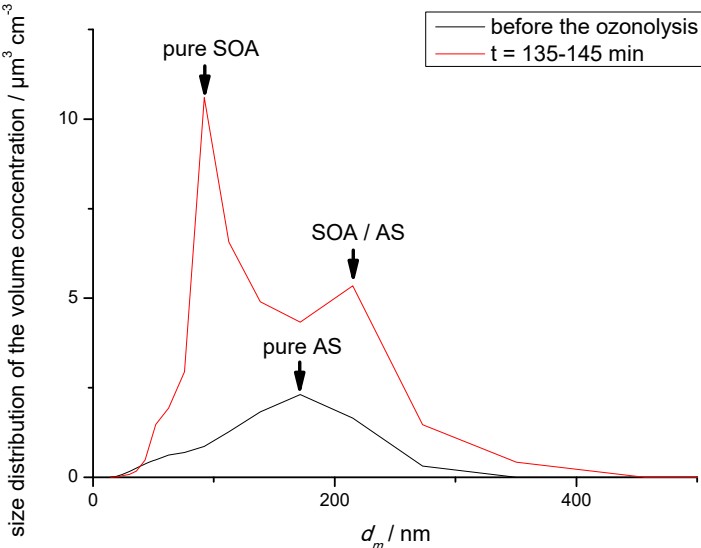


**Figure 4:** Particle volume size distribution before and ten minutes after the addition of ozone.


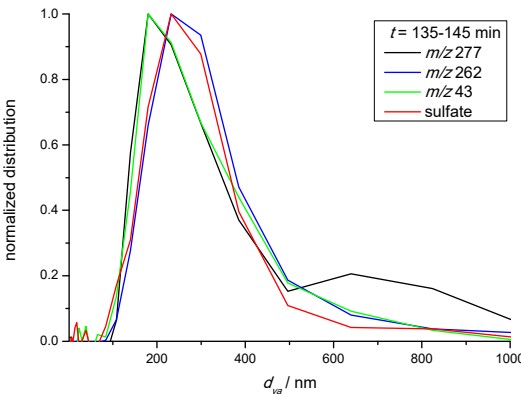

**Figure 5:** Size distribution of individual particle phase compounds (TPPO, TPP, SOA, sulfate) within the first 10 minutes after the addition of ozone.



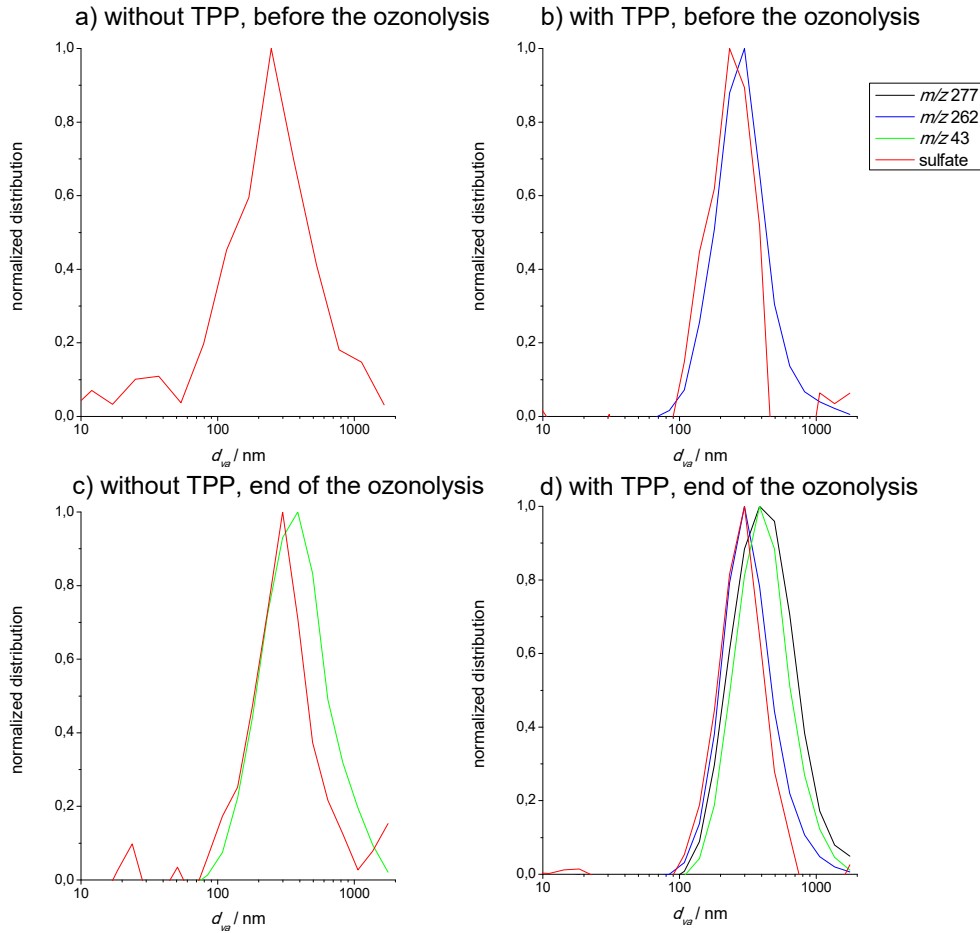

**Figure 6:** Size distribution of *m/z* 277 (black = TPPO), *m/z* 262 (blue = residual TPP), *m/z* 43 (green = SOA), sulfate (red) for following periods: Before the start of the ozonolysis a) without and b) with adding TPP, at the end (185-265 min) of the ozonolysis c) without and d) with adding TPP. All distributions were normalized to their respective maximum.





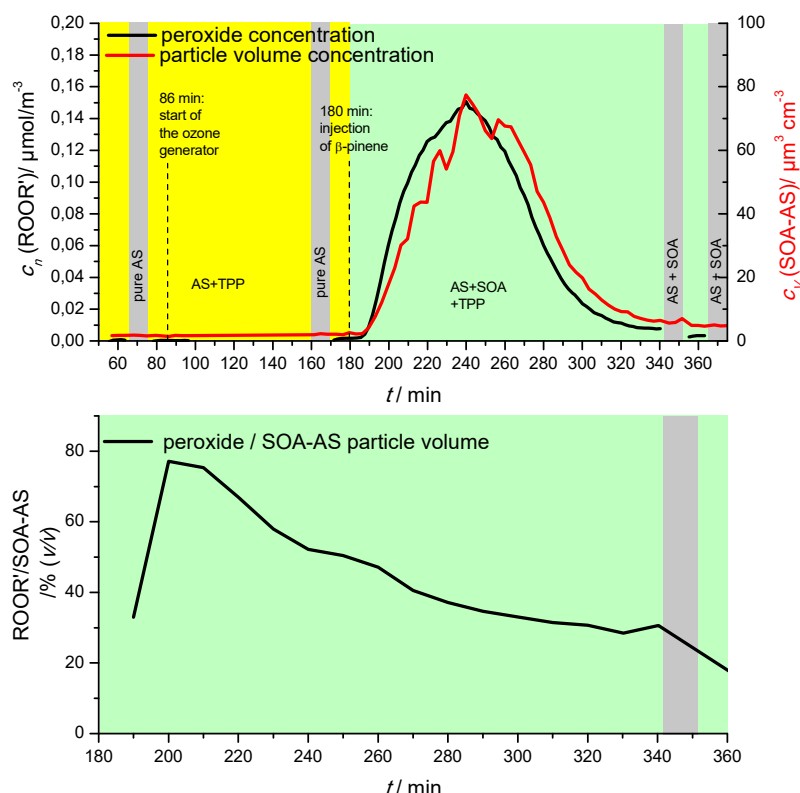


**Figure 7:** Ozonolysis of β-pinene in the presence of AS aerosol. Time series of a) peroxide (black), particle volume concentration (red) and

b) relative contribution of peroxide to the total aerosol volume. Ozone was added at 86 min, β-pinene at 180 min. Averages molar masses of

250 g mol$^{-1}$ and 300 g mol$^{-1}$ for SOA and peroxides were assumed, respectively.



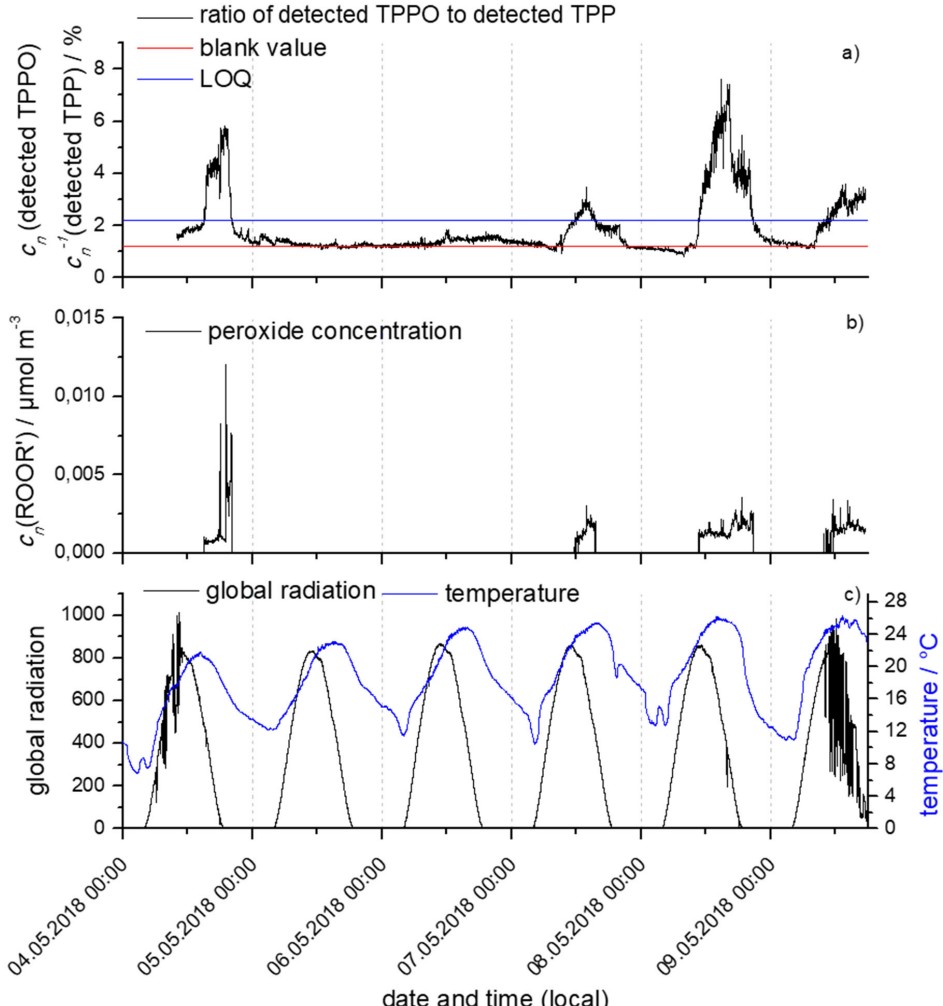

**Figure 8:** a) Time series of the relative amount of detected TPPO to detected TPP, LOQ and blank value, b) diurnal variation of the estimated peroxide concentration and c) time series of the global radiation and temperature.

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
