# Peer review of "Application of time-of-flight aerosol mass spectrometry for the realtime measurement of particle phase organic peroxides: An online redox derivatization – aerosol mass spectrometer (ORD-AMS)"

_Atmospheric Measurement Techniques, 2019_

## Referee Comment (RC1) · Anonymous Referee #1 · 2 Dec 2019

Weloe and Hoffman present a new technique to measure and quantify organic peroxides in secondary organic aerosol (SOA) with the Aerodyne Aerosol Mass Spectrometer (AMS). They flow the sample (both laboratory generated and ambient SOA through a reaction volume, where the organic peroxides in SOA reacts with triphenylphosphine (TPP) to form triphenylphosphine oxide (TPPO), and the AMS measures the TPPO to quantify the organic peroxides in SOA. As the AMS uses hard ionization (70 eV electron beam to ionize the gases), speciation of SOA compounds is typically limited and

challenging. This novel technique provides a way to allow the AMS to start identifying classes of compounds in SOA. Also, this provides a potentially useful method in minimizing sample handling, reducing potential experimental concerns (e.g., reactions on filters prior to sample handling). However, there are measurements and experiments that are currently missing that limits the current understanding, validity, and limits of this technique. After revising, this paper will fit into the scope of AMT and will be of great benefit for the AMS and analytical community.

Major Concerns: (1) The biggest thing missing from this paper is that the authors did not run any organic peroxide standards to investigate this technique. Running standards of a couple organic peroxides that are atomized into aerosol to investigate the percent conversion/yield of TPP to TPPO in the AMS would be beneficial in understanding the quantification. For example, when TPP is used in laboratory, it takes ∼30 minutes to convert the organic peroxide to TPPO (e.g., Chiba et al., 1989; Aimanant & Ziemann, 2013). However, the authors don't discuss the reaction time/residence time, if TPP is always in excess, etc.

(2) Further, experiments that indicate the percent conversion/yield in solid vs liquid SOA and mixtures, both with known standards and amounts, is needed, to understand the quantification. As SOA can range from solid to liquid in the atmosphere and chamber studies, does the percent conversion remain constant or does it increase as the SOA becomes more liquid? Further, does the percent conversion remain constant from a purely organic peroxide atomized aerosol to a mixture of different percents of organic peroxides with other types of organics?

(3) For both experiments, showing a comparison of the AMS quantification vs another method (e.g., iodometric) (e.g., Epstein et al., 2014) would provide further confidence in the quantification of organic peroxides in the AMS through this technique (esp. since this is a new technique, a benchmark vs a well established technique is needed for further proof-of-concept and understanding of the quantification).

[Figure]

(4) Are there potential reactions with other functional groups that may lead to mis-quantification as an organic peroxide? E.g., if there is aerosol with hydrogen peroxide (Chellmani & Suresh, 1988), which occurs in ambient air or epoxides (Sonnet & Oliver, 1976), does this lead to a positive bias in the TPP conversion to TPPO and in turn a positive bias in the amount of organic peroxides prescribed by this method?

Minor concerns: There are many grammatical issues throughout the text, making some sentences very difficult to understand. Please revise.

Check references. Some references are listed twice (e.g., Canagaratna et al., 2007; Mertes et al., 2012).

For the TPP + ROOR' –> TPPO + ROR' figure, could the authors show a more explicit arrow pushing/mechanism, as it is currently not very clear how oxygen is transferred to TPP and the organic keeps both R groups.

Line 116: What is CSTR?

Line 117: What is the residence time for the flow reactor?

Line 120 - 122: Is the ammonium sulfate seed charged and size selected?

Section 2.4: What was the RH & temperature for the experiment?

Line 186 - 187: Could you provide some examples for the reactions, instead of A+B–> C and C–>D?

Line 197: Instead of assuming 1.2 g cm-3, why did the authors not take the parameterization from O/C and H/C ratios (Kuwata et al., 2011).

For the PToF figures (Fig. 5 & 6) and starting at line 214, please include total OA with m/z 43.

As the m/z 44 vs m/z 43 triangle plot or H/C vs O/C plot has been used to signify oxidized and potentially organic peroxides (e.g., Ng et al., 2010; 2011), how does that

appear for your experiments (colored by peroxide/SOA, for example).

References:

Aimanant & Ziemann, Aerosol Sci. Technol., doi:10.1080/02786826.2013.773579, 2013

Chellmani & Suresh, Reaction Kinetic & Catalysis Letters, doi:10.1007/BF02062106, 1988

Chiba et al., J. American Oil Chem. Soc., doi:10.1007/BF02636182, 1989

Epstein et al., Environ. Sci. Technol., doi:10.1021/es502350u, 2014

Kuwata et al., Environ. Sci. Technol., doi:10.1021/es202525q, 2011

Ng et al., Atmos. Chem. Phys., www.atmos-chem-phys.net/10/4625/2010, 2010

Ng et al., Atmos. Chem. Phys., www.atmos-chem-phys.net/11/6465/2011, 2011

Sonnet & Oliver, J. Org. Chem., doi:10.1021/jo00882a015, 1976
* * *

---

## Referee Comment (RC2) · Anonymous Referee #2 · 7 Dec 2019

Referee comments for:

Weloe, M. and Hoffmann, T.: Application of time-of-flight aerosol mass spectrometry for the real-time measurement of particle phase organic peroxides: An online redox derivatization – aerosol mass spectrometer (ORD-AMS), Atmos. Meas. Tech. Discuss., https://doi.org/10.5194/amt-2019-406, in review, 2019.

General comments:

[Figure]

Weloe and Hoffmann describe an online derivatization technique for the measurement of peroxides in aerosol particles in real-time using an aerosol mass spectrometer (AMS). The technique involves using the reaction of triphenylphosphine (TPP) with peroxides, generating triphenylphosphine oxide (TPPO) which is measurable by the AMS. The authors use the TPP/TPPO system to measure the particle-phase peroxides formed in secondary organic aerosol (SOA) chamber experiments of alpha- and beta-pinene ozonolysis. Peroxide concentrations are measured to increase with the onset of SOA formation, then decrease over time; however, these results are not validated with other techniques or standards. Since peroxides are thought to be important species for SOA formation and most techniques measuring peroxides require offline analysis, the real-time measurement of such species would be of great interest to the atmospheric chemistry community. Despite the potential utility of the described technique, the paper contains several large unanswered questions and deficiencies that should be addressed before publication. Unless major changes are made and additional work is done, this cannot be considered a quantitative technique (as stated in the abstract). Finally, the manuscript should be proofread for English grammar before resubmission. Scientific questions and grammatical corrections are listed below.

Specific comments:

One major question is what is the extent of reaction/derivatization between SOA and TPP? The residence time in the condenser/reaction volume is not given, but likely impacts the extent of derivatization. What tests were done to determine whether or not the aerosol was fully reacted with TPP? I do not see any mention of varying the reaction time and how this impacts the observed concentration of TPPO. At the very least, the diffusion of TPP in an organic matrix could be estimated to determine if the reaction time is long enough to get full mixing between a typical organic particle and the derivatization agent. In addition, it's possible that reaction of TPP with semisolid organic particles might be limited to the particle surface, thus limiting ability of the technique to give quantitative peroxide concentrations. Please comment on the potential effect of

high viscosity/semisolid particles and their reaction with TPP.

The other major issue with the manuscript is the lack of experiments using organic peroxide standards to help determine the efficacy of the derivatization technique. While a suitable standard may be difficult to find, there are commercially available organic peroxides which could be used. Some combination of validation using standards or offline techniques to compare measured peroxide concentrations (from TPP/TPPO system) should be used to validate the technique.

The ionization efficiency of TPP and TPPO should be stated in the main text earlier, potential at the end of section 2.3 or in section 3.2.

"CSTR" needs to be defined in line 116.

How was the ozone concentration estimated (line 128)?

What is meant by "regular MS" (line 132)? Does this refer to EI-MS?

If a high resolution AMS was used, why use unit mass resolution peaks (e.g., m/z 277 for TPPO) rather than the high-resolution m/z values or identified ions?

In section 3.3, it is stated that the experiments showed a constant background contribution of TPPO from TPP; however, these values vary in the SI. What is the impact on calculated peroxide concentrations when experiment specific corrections are used?

During SOA experiments, there are periods where TPP is not added. Please describe how this is accomplished? Also, in Figure 3 (a and b), why are there gaps in the data (excluding the grey bars)? In Fig. 3b, why is the volume so variable?

In Figure 3a, what causes the increase in detected TPP when SOA is formed? Is it because there is additional surface area for the TPP to condense onto? Or is it due to some increase in collection efficiency?

In the Fig. 3 caption, the time resolution is given as 10 minutes, but the abstract says 1-2 minutes.

Why use SMPS volume to quantify the SOA rather than the AMS? And why quantify peroxide concentrations in terms of volume at all, instead of mass?

It isn't clear what corrections are done to the data, if any. For example, is the data corrected for dilution in the chamber or AMS collection efficiency?

What systems were studied in the references given in lines 201 and 202 (Epstein et al. 2014, Li et al. 2016, Mertes et al. 2012)? It's not clear how these studies are related to the manuscript.

Does the formation of SOA interfere with the quantification of TPP/TPPO in the AMS mass spectra? AMS mass spectra of SOA + TPP/TPPO should be shown either in the main manuscript or the SI.

Does all of the TPPO that is formed from the SOA + TPP reaction stay in the particle phase? Any partitioning of TPPO to the gas phase would limit the quantification of peroxide.

In Figure 5, only m/z 277 looks to be bimodal, not the organic or sulfate as stated in the manuscript (line 217). I also don't find the inclusion of all of the size distribution data to be particularly useful. Some of these figures and discussion could be moved to the SI.

In line 263, ozone is excluded as a direct oxidant for TPP. This should be mentioned much earlier in the manuscript given that it is an obvious question that clouds the interpretation of the preceding results.

Technical corrections:

Line 31, remove "however"

Line 73 should read "and use this"

Line 85 "carrier gas to deliver"

Line 99 "functions as a detector"

Line 100 "the beam passes"

Line 121 "was nebulized"

Line 140 "background signal has to be"

Line 191 "In any case, the"

Line 195 "As seen in"

Line 202 "Mertes and coworkers"

Line 240 "diameter depends on"

Line 253 "increased rapidly with the formation of SOA" or something to this effect

Line 258 "SOA concentration supports the concept"

Line 276 says only peroxides are detected during weekdays, but the weekend days are listed in parentheses

———————————————

---

## Author Comment (AC1) · 30 Apr 2020

**AMT-2019-406- Reviewer Comment 1**

Major Concerns:

(1) The biggest thing missing from this paper is that the authors did not run any organic peroxide standards to investigate this technique. Running standards of a couple organic peroxides that are atomized into aerosol to investigate the percent conversion/yield of TPP to TPPO in the AMS would be beneficial in understanding the quantification. For example, when TPP is used in laboratory, it takes _30 minutes to convert the organic peroxide to TPPO (e.g., Chiba et al., 1989; Aimanant & Ziemann, 2013). However, the authors don't discuss the reaction time/residence time, if TPP is always in excess, etc.

Answer: The reviewer is right, and we admit that this is a weakness of the manuscript. In fact, however, we performed experiments with several commercially available organic peroxides, e.g. lauryl peroxide or benzoyl peroxide, and the qualitative results were as expected, i.e. peroxide compounds gave a clear signal, non-peroxide compounds did not (when background correction was applied). One of the problems with the peroxide standards studied is probably related to a substantial change in volatility after the reaction with TPP, which is likely a result of the breaking of the molecules into smaller units, e.g. the formation of benzoic acid from benzoyl peroxide. The changing aerosol mass also changed the partitioning of the analytes, which made it difficult to quantitatively determine the reference compounds. In fact, the same behaviour can be expected for structurally similar R-O-O-R peroxides in the atmosphere, and it can probably be said that the method developed is semi-quantitative for such compounds. Nevertheless, it can be assumed that most peroxides in the ambient atmosphere are hydroperoxides. Although there is certainly a vapour pressure difference between the hydroperoxides and the corresponding OH functionality after the reaction with TPP (R-O-O-H -> R-O-H) (Compernolle et al., 2010, Atmos. Chem. Phys.), the difference is not of several orders of magnitude as in the case of the standards used for this study. Therefore, we used the standards for the general proof of feasibility of the method and not for calibration. To make this limitation of the method clear, we have included an additional paragraph in the manuscript (section 3.1):
" Experiments with peroxides (R-O-O-R) as reference compounds were also performed, using commercially available organic peroxides. e.g. lauryl peroxide and benzoyl peroxide. While these experiments clearly showed the formation of TPPO from TPP, the use of these compounds for quantification was not possible, since the peroxide standards studied undergo a substantial change in volatility after the reaction with TPP. Therefore, lauryl peroxide and benzoyl peroxide were used to demonstrate the general proof of feasibility of the method, however, not for calibration. "

Reaction time and concentration ratios of analytes and reactants are now also discussed and mentioned in the revised manuscript text.

(2) Further, experiments that indicate the percent conversion/yield in solid vs liquid SOA and mixtures, both with known standards and amounts, is needed, to understand the quantification. As SOA can range from solid to liquid in the atmosphere and chamber studies, does the percent conversion remain constant or does it increase as the SOA becomes more liquid? Further, does the percent conversion remain constant from a purely organic peroxide atomized aerosol to a mixture of different percents of organic peroxides with other types of organics?

Answer: Here too the reviewer has a point. SOA is a complicated matrix, and several experimental/environmental parameters can affect the results. Therefore the phase state of SOA can also influence the peroxide measurements, as discussed in the manuscript. However, the main advantage of the presented method is the high time resolution, which to our knowledge is not achieved by other techniques. Therefore, we believe that the developed method should be made

available to other researchers, although not all possible effects on the measurement results have been investigated in detail. It should also be mentioned that the relative humidity was always above 60% and the temperature between 22 and 25°C. Consequently, we expect the SOA to be liquid for all experiments performed. All aspects discussed here are now also discussed in the revised manuscript:

"Another aspect of SOA experiments which include chemistry, as in the case of peroxide-TPP reaction, that have to be considered is the question of the phase state of SOA. According to Koop et al. (2011) SOA is expected to be a liquid under the condition used for our experiments (22-25 °C, r.h. 60%). Also the AS seed particles can be expected to behave liquid-like (Matthew et al., 2008). As a consequence the diffusion time of TPP within the particles should be approximately one second (Koop et al., 2011), which is much shorter than the residence time of about 35 seconds of the aerosol in the ORD-setup (Figure 1). In conclusion, based on the agreement of the peroxide/SOA ratio at a later stage of the experiments measured in this work and the results of former studies, the estimated diffusion times and the excess of the reactant TPP to SOA, a quantitative reaction of TPP with the particle phase peroxides to TPPO can be assumed. However, in future investigations, the method should be revalidated using as aerosol particles composed of pure single component peroxides."

(3) For both experiments, showing a comparison of the AMS quantification vs another method (e.g., iodometric) (e.g., Epstein et al., 2014) would provide further confidence in the quantification of organic peroxides in the AMS through this technique (esp. Since this is a new technique, a benchmark vs a well established technique is needed for further proof-of-concept and understanding of the quantification).

The difficulty here lies in the comparability with other methods (e.g. iodometry), since here too the time resolution is completely different. Only at the end of the experimental runs, when the peroxide concentration becomes stable, such comparisons can be made. Exactly this is described in the manuscript, but not carried out by the authors themselves, but by comparison with the results of other groups who used the same chemical systems ($\alpha$-, ß-pinene/ozone). An additional sentence in the revised manuscript points out the detailed procedure. However, we agree with the reviewer that in future studies with the described method additional simultaneous measurements would be helpful.

(4) Are there potential reactions with other functional groups that may lead to misquantification as an organic peroxide? E.g., if there is aerosol with hydrogen peroxide (Chellmani & Suresh, 1988), which occurs in ambient air or epoxides (Sonnet & Oliver, 1976), does this lead to a positive bias in the TPP conversion to TPPO and in turn a positive bias in the amount of organic peroxides prescribed by this method?

The reviewer is right. Hydrogen peroxide in the particle phase would also be measured with the method presented. However, this will be the case with most peroxide measurement techniques (e.g. iodometry) and could only be avoided by a combination of enzymatic methods or the use of separation techniques (e.g. LC). In both cases the essential advantage - the high time resolution - of the presented technique would be lost. Although a clear distinction between hydrogen peroxide and organic peroxides would be desirable, a close chemical connection between the two analytes can always be expected, e.g. the hydrolysis of organic peroxides forming $H_2O_2$. The second reference mentioned by the reviewer (Sonnet & Oliver, 1976) investigated a different chemical system (reaction of triphenylphosphine dihalides with epoxides).

Minor concerns: There are many grammatical issues throughout the text, making some sentences very difficult to understand. Please revise.

We have tried to fix as many errors as possible

Check references. Some references are listed twice (e.g., Canagaratna et al., 2007; Mertes et al., 2012).

Corrected

For the TPP + ROOR' –> TPPO + ROR' figure, could the authors show a more explicit arrow pushing/mechanism, as it is currently not very clear how oxygen is transferred to TPP and the organic keeps both R groups.

We introduced now more information about the possible reaction mechanism and show additional reaction equations of the TPP + ROOR' reaction.

Line 116: What is CSTR?

Explained now in the text : continuous flow tank reactor

Line 117: What is the residence time for the flow reactor?

The values are now given in the edited manuscript: SOA-Reaktor: 13,5 min; TPP-Reaktor: 35 s

Line 120 - 122: Is the ammonium sulfate seed charged and size selected?
The AS seed particles was not dried but not size selected as described in the text.

Section 2.4: What was the RH & temperature for the experiment?

These data are given in the text now.

Line 186 - 187: Could you provide some examples for the reactions, instead of A+B–> C and C–>D?

Several reactions are listed now:
"…..reactions with alcohols or aldehydes forming aldehydes or carboxylic acids, or hydrolysis followed by the loss of H2O2 (Zhao et al., 2018; Claflin et al., 2018; Epstein et al., 2014)"

Line 197: Instead of assuming 1.2 g cm-3, why did the authors not take the parameterization

from O/C and H/C ratios (Kuwata et al., 2011).

The density values are quite similar, 1.19 g/cm³ for H:C = 1.64 and O:C = 0.41., ß-pinene: 1.24 g/cm³ for H:C = 1.47 and O:C = 0.40.

For the PToF figures (Fig. 5 & 6) and starting at line 214, please include total OA with m/z 43.

We could add a figure such as the figure below, indicating that m/z 43 is representative for OA. However, we are not convinced that an additional figure is necessary.

[Figure]

As the m/z 44 vs m/z 43 triangle plot or H/C vs O/C plot has been used to signify oxidized and potentially organic peroxides (e.g., Ng et al., 2010; 2011), how does that appear for your experiments

This is certainly an interesting point for discussion, however, beyond the scope pf the manuscript. We will pick that up in the upcoming paper about the presented technique.

---

## Author Comment (AC2) · 30 Apr 2020

**AMT-2019-406-Reviewer Comment 2**

Specific comments:

One major question is what is the extent of reaction/derivatization between SOA and TPP? The residence time in the condenser/reaction volume is not given, but likely impacts the extent of derivatization. What tests were done to determine whether or not the aerosol was fully reacted with TPP? I do not see any mention of varying the reaction time and how this impacts the observed concentration of TPPO. At the very least, the diffusion of TPP in an organic matrix could be estimated to determine if the reaction time is long enough to get full mixing between a typical organic particle and the derivatization agent. In addition, it's possible that reaction of TPP with semisolid organic particles might be limited to the particle surface, thus limiting ability of the technique to give quantitative peroxide concentrations. Please comment on the potential effect of high viscosity/semisolid particles and their reaction with TPP.

The other major issue with the manuscript is the lack of experiments using organic peroxide standards to help determine the efficacy of the derivatization technique. While a suitable standard may be difficult to find, there are commercially available organic peroxides which could be used. Some combination of validation using standards or offline techniques to compare measured peroxide concentrations (from TPP/TPPO system) should be used to validate the technique.

Answer: The reviewer is right and actually using the same arguments as reviewer 1. Therefore the answer to reviewer 2 is identical:
The reviewer is right we admit that this is a weakness of the manuscript. In fact, however, we performed experiments with several commercially available organic peroxides, e.g. lauryl peroxide or benzoyl peroxide, and the qualitative results were as expected, i.e. peroxide compounds gave a clear signal, non-peroxide compounds did not (when background correction was applied). One of the problems with the peroxide standards studied is probably related to a substantial change in volatility after the reaction with TPP, which is likely a result of the breaking of the molecules into smaller units, e.g. the formation of benzoic acid from benzoyl peroxide. The changing aerosol mass also changed the partitioning of the analytes, which made it difficult to quantitatively determine the reference compounds. In fact, the same behaviour can be expected for structurally similar R-O-O-R peroxides in the atmosphere, and it can probably be said that the method developed is semi-quantitative for such compounds. Nevertheless, it can be assumed that most peroxides in the ambient atmosphere are hydroperoxides. Although there is certainly a vapour pressure difference between the hydroperoxides and the corresponding OH functionality after the reaction with TPP (R-O-O-H -> R-O-H) (Compernolle et al., 2010, Atmos. Chem. Phys.), the difference is not of several orders of magnitude as in the case of the standards used for this study. Therefore, we used the standards for the general proof of feasibility of the method and not for calibration. To make this limitation of the method clear, we have included an additional paragraph in the manuscript (section 3.1):
" Experiments with peroxides (R-O-O-R) as reference compounds were also performed, using commercially available organic peroxides. e.g. lauryl peroxide and benzoyl peroxide. While these experiments clearly showed the formation of TPPO from TPP, the use of these compounds for quantification was not possible, since the peroxide standards studied undergo a substantial change in volatility after the reaction with TPP. Therefore, lauryl peroxide and benzoyl peroxide were used to demonstrate the general proof of feasibility of the method, however, not for calibration. "

Reaction time and concentration ratios of analytes and reactants are now also discussed and mentioned in the revised manuscript text.

About the viscosity of OA: Here too the reviewer has a point. SOA is a complicated matrix, and several experimental/environmental parameters can affect the results. Therefore the phase state of SOA can also influence the peroxide measurements, as discussed in the manuscript. However, the main advantage of the presented method is the high time resolution, which to our knowledge is not

achieved by other techniques. Therefore, we believe that the developed method should be made available to other researchers, although not all possible effects on the measurement results have been investigated in detail. It should also be mentioned that the relative humidity was always above 60% and the temperature between 22 and 25°C. Consequently, we expect the SOA to be liquid for all experiments performed. All aspects discussed here are now also discussed in the revised manuscript:

"Another aspect of SOA experiments which include chemistry, as in the case of peroxide-TPP reaction, that have to be considered is the question of the phase state of SOA. According to Koop et al. (2011) SOA is expected to be a liquid under the condition used for our experiments (22-25 °C, r.h. 60%). Also the AS seed particles can be expected to behave liquid-like (Matthew et al., 2008). As a consequence the diffusion time of TPP within the particles should be approximately one second (Koop et al., 2011), which is much shorter than the residence time of about 35 seconds of the aerosol in the ORD-setup (Figure 1). In conclusion, based on the agreement of the peroxide/SOA ratio at a later stage of the experiments measured in this work and the results of former studies, the estimated diffusion times and the excess of the reactant TPP to SOA, a quantitative reaction of TPP with the particle phase peroxides to TPPO can be assumed. However, in future investigations, the method should be revalidated using as aerosol particles composed of pure single component peroxides."

"CSTR" needs to be defined in line 116.

Is defined now in the text: continuous flow tank reactor

How was the ozone concentration estimated (line 128)?

The ozone monitor is calibrated using wet chemical methods on a regular basis.

What is meant by "regular MS" (line 132)? Does this refer to EI-MS?

Regular MS refers to the ionization process in the AMS compared to a conventional EI mass spectrometer. Due to rapid evaporation, the compounds fragment more strongly in the AMS. The mass spectrum of the AMS shows the same fragments, but with a different intensity distribution than the reference spectra.

If a high resolution AMS was used, why use unit mass resolution peaks (e.g., m/z 277 for TPPO) rather than the high-resolution m/z values or identified ions?

No signals have been detected at mass 277 except for the [M-1]+ ion of TPPO. The signal of the UMR peak is therefore as high as that of the HR peak. At mass 262, only signals from TPP were detected, so HR is not necessary here either. Since the evaluation by UMR is easier and faster, HR peaks were omitted.

[Figure]

In section 3.3, it is stated that the experiments showed a constant background contribution of TPPO from TPP; however, these values vary in the SI. What is the impact on calculated peroxide concentrations when experiment specific corrections are used?

The background was 1.08% in the presented experiment for a-pinene and 1.33% for b-pinene. If one would take these values, the peroxide concentration would increase by 2% or decrease by 1 to 18%. The smaller the TPPO to TPP ratio, the greater the differences. Therefore, only data above the limit of determination of 2.25% were evaluated. As the origin of the background TPPO is unknown, the TPPO/TPP ratio was averaged.

During SOA experiments, there are periods where TPP is not added. Please describe how this is accomplished? Also, in Figure 3 (a and b), why are there gaps in the data (excluding the grey bars)?

To measure SOA with the AMS, T-piece 1 was connected directly to T-piece 2 without the ORD setup. In order to determine the peroxide content again, the ORD setup was reconnected. During this phase, the data acquisition of the AMS and SMPS was interrupted and continued only after the particle concentrations at the CPC reach a constant value again. Therefore, there are the gaps in Fig 3.

In Fig. 3b, why is the volume so variable?

Technical problems with the SMPS, probably because the scanning time per diameter was chosen very short in order to achieve the shortest possible total scanning time of 200 s.

In Figure 3a, what causes the increase in detected TPP when SOA is formed? Is it because there is additional surface area for the TPP to condense onto? Or is it due to some increase in collection efficiency?

[Figure]

TPP is only detected by AMS when it condenses onto particles. The more particles enter the condenser, the more TPP is detected. According to Figures 4 and 5. mainly new SOA particles are formed, but growth of the seed aerosol can also be observed. Since wall losses are size-dependent and for uncharged particles the minimum for the losses is about 150 nm, the enlargement of the pure AS particles from about 33 nm to over 100 nm should lead to less particle deposition in the chamber. The AMS ultimately only measures the particle concentration before the SMPS and ORD-AMS branches.

[Figure]

In the Fig. 3 caption, the time resolution is given as 10 minutes, but the abstract says 1-2 minutes.

The time resolutions for AMS and SMPS were 120 s and 200 s respectively. To compare the peroxide concentration with the volume concentration, the values were averaged to 600 s (10 min). It refers only to figure c).

Why use SMPS volume to quantify the SOA rather than the AMS? And why quantify peroxide concentrations in terms of volume at all, instead of mass?

In ORD-AMS mode, no SOA and AS concentrations can be determined because fragments of TPP and TPPO match SOA and AS fragments. Especially since TPP makes up a large part of the aerosol, its signals are very high compared to the signals of other compounds. The subtraction of TPP signals from the rest of the spectrum is therefore prone to errors. Correction: Molar SOA concentration is calculated assuming constant density.

It isn't clear what corrections are done to the data, if any. For example, is the data corrected for dilution in the chamber or AMS collection efficiency?

The addition of TPP in an N2 stream to the aerosol results in a dilution of the aerosol. The dilution and other particle losses during the transmission from T-piece 1 to T-piece 2 are corrected by the CPC. For this purpose, the particle concentration is measured when switching between ORD-AMS and normal AMS mode. The dilution can be corrected from the ratio. For a-pinene the dilution was 0.7 and for b-pinene 0.6. Particle losses in the chamber and the collection efficency of the AMS were not taken into account.

What systems were studied in the references given in lines 201 and 202 (Epstein et al. 2014, Li et al. 2016, Mertes et al. 2012)? It's not clear how these studies are related to the manuscript.

In the references the ozonolysis of a-pinene was also examined for peroxide content. The aerosol was collected on filters or impactors and the peroxide content was analysed iodometrically. However, several parameters such as the terpene/ozone ratio, temperature and humidity were different. Nevertheless, the peroxide content were observed to be between 34 and 12%. Our chamber conditions best match those of Li et al (2016). There, a yield of 21% was determined, which was also found in their own experiments. The effect of seed aerosol (5%) on the peroxide yield should be low for a-pinene.

Does the formation of SOA interfere with the quantification of TPP/TPPO in the AMS mass spectra? AMS mass spectra of SOA + TPP/TPPO should be shown either in the main manuscript or the SI.

According to (DeCarlo et al. 2006) the ionization of the compounds after evaporation from the particles in high vacuum should take place independently. Thus, quantification should be independent of the aerosol matrix.  As can be seen from the comparison of pure  SOA to

SOA+TPP+TPPO, there is no significant overlap of the signal at m/z 262 and m/z 277 with SOA signals, therefore these signals can be used to quantify TPP and TPPO.

[Figure]

*Pinene ozonolysis products with TPP*

[Figure]

*Pinene ozonolysis products without TPP*

Does all of the TPPO that is formed from the SOA + TPP reaction stay in the particle phase? Any partitioning of TPPO to the gas phase would limit the quantification of peroxide.

A valid point. However, according to the manufacturer, the vapour pressure at 50 °C is less than 1 hPa and the boiling point is over 360 °C. Therefore, we assume a non-volatile substance although we cannot completely rule out this possibility.

In Figure 5, only m/z 277 looks to be bimodal, not the organic or sulfate as stated in the manuscript (line 217).I also don't find the inclusion of all of the size distribution data to be particularly useful. Some of these figures and discussion could be moved to the SI.

We prefer to have the figures in the manuscript.

In line 263, ozone is excluded as a direct oxidant for TPP. This should be mentioned much earlier in the manuscript given that it is an obvious question that clouds the interpretation of the preceding results.

Technical corrections:
Line 31, remove "however"
Line 73 should read "and use this"
Line 85 "carrier gas to deliver"
Line 99 "functions as a detector"
Line 100 "the beam passes"

Line 121 "was nebulized"
Line 140 "background signal has to be"
Line 191 "In any case, the"
Line 195 "As seen in"
Line 202 "Mertes and coworkers"
Line 240 "diameter depends on"
Line 253 "increased rapidly with the formation of SOA" or something to this effect
Line 258 "SOA concentration supports the concept"
Line 276 says only peroxides are detected during weekdays, but the weekend days are listed in parentheses

all done

Edited figures:

[Figure]

Figure 1

[Figure]

Figure 3

[Figure]

Figure 7

---

## Author Response (AR2)

**Point-by-point response to editor and the reviewer comments:**

**AMT-2019-406**

Associate Editor Decision: Publish subject to minor revisions (review by editor) (29 Jun 2020) by Charles Brock

Comments to the Author:
Thanks for your revisions to the original manuscript that have largely satisfied the reviewers' concerns. Please address the remaining minor issues they highlight (combined below). I support Referee #2's suggestion to describe the method as "semi-quantitative" (but very promising) at this point. In addition, please carefully go through the references and ensure that the formatting matches Copernicus requirements. For example, sometimes journals are fully spelled out rather than abbreviated, sometimes article titles are capitalized and sometimes not, etc. (see Bianchi et al., Donahue et al., Brueggemann et al., 2015 for examples). This is an unfortunate consequence of EndNote-type software--it never gets the formatting quite right.

Answer: Thank you for the suggestions of the editor and reviewers and the generally positive responses. As you can see below, we have followed the suggestions to a large extent and hope to present a better manuscript indeed. So we now use the term semi-quantitative and also the rightly noted non-uniform citation style has been adapted.

Referee #1:

As a proof-of-concept paper and introduction of a new technique, I recommend publishing the paper after the authors address the concerns presented below.

Page 4, line 134: I think the wrong figure maybe referenced.
Answer: We have deleted the reference because it was misleading at this point.

Page 5, line 183: "according to the reaction equation the same as the concentration" is unclear what the authors mean
Answer: The sentence was reworded and now reads: "Based on the stoichiometry of the reaction the molar concentration cn of peroxides is the same as the concentration of TPPO and can be calculated from the signal at m/z 277 as:"

Page 6, line 192: "As mentioned above also TPP particles" is unclear what the authors mean
Answer: We have added a reference that should make the statement understandable now: "As mentioned above (see 3.1) also TPP particles …."

Page 6, line 197: if the error is only shown to the tenth unit (0.1) then the background contribution should be similar (1.2)
Answer: Has been corrected.

Page 6, line 201 (Eq. 10): It's unclear what the 0.012125 came from or what it represents
Answer: The "0.012125" refers to the 1.2% background contribution. To make this more understandable the text is now slightly changed and reads: "These experiments showed a constant background contribution of 1.2 (+/-0.1) % TPPO resulting from the conversion of TPP (m/z 262 => Cn (detected TPP))."

Page 6, line 211 - 212: Please mention that you will explain later why TPP rose quickly once ozone was introduced. Without saying you explain why, it is a shock.
Answer: Yes - could be. Therefore the text at this point now reads as follows: "Why the TPP signal also rises will be explained below. However, peroxides were only detected when SOA formation began, as indicated by the time series of peroxide concentration (Fig. 3b, black) and SOA mass concentration (Fig. 3b, blue line)."

Page 7, line 265: It should be "since TPP also moves"
Answer: Has been corrected.

Figure 6: Since you are comparing the different mass distributions, it would be use to at least mark where the maxima was from the prior panel. Since it is in log scale, it's hard to mentally line up the changes in diameter that is described in the text.
Answer: We have now added the x-axis to the top of the graph to make the position of the maxima easier.

Supplement: I appreciate the authors using Kuwata et al. to estimate the SOA density; however, I do not recall the authors referencing they did this in the main paper. Instead, the only reference for density is on page 7, line 233, where they state a value that is sometimes assumed as it was measured in other studies. Please reference the supplement.
Answer: Kuwata et al. 2012 is now cited in the main text.

Referee #2:
- "to detect" rather than "to detected" (line 12).
Answer: Has been corrected.

- I would prefer that the technique demonstrated be described as "semi-quantitative" rather than "quantitative" (line 18).
Answer: Has been corrected.

[revised manuscript text omitted]